# The Interactions between Nanoparticles and the Innate Immune System from a Nanotechnologist Perspective

**DOI:** 10.3390/nano11112991

**Published:** 2021-11-06

**Authors:** Lena M. Ernst, Eudald Casals, Paola Italiani, Diana Boraschi, Victor Puntes

**Affiliations:** 1Vall d’Hebron Research Institute (VHIR), 08035 Barcelona, Spain; lena.montana@vhir.org; 2School of Biotechnology and Health Sciences, Wuyi University, Jiangmen 529020, China; wyuchemecm@126.com; 3Institute of Protein Biochemistry and Cell Biology (IBBC), National Research Council (CNR), 80131 Napoli, Italy; paola.italiani@ibbc.cnr.it (P.I.); diana.boraschi@ibbc.cnr.it (D.B.); 4Shenzhen Institute of Advanced Technology (SIAT), Chinese Academy of Sciences (CAS), Shenzhen 518055, China; 5Stazione Zoologica Anton Dohrn, 80121 Napoli, Italy; 6Institut Català de Nanociència i Nanotecnologia (ICN2), CSIC and The Barcelona Institute of Science and Technology (BIST), Campus UAB, 08193 Barcelona, Spain; 7Institució Catalana de Recerca I Estudis Avançats (ICREA), 08010 Barcelona, Spain

**Keywords:** nanoparticles, immune system, innate immunity, inflammation, tolerance

## Abstract

The immune system contributes to maintaining the body’s functional integrity through its two main functions: recognizing and destroying foreign external agents (invading microorganisms) and identifying and eliminating senescent cells and damaged or abnormal endogenous entities (such as cellular debris or misfolded/degraded proteins). Accordingly, the immune system can detect molecular and cellular structures with a spatial resolution of a few nm, which allows for detecting molecular patterns expressed in a great variety of pathogens, including viral and bacterial proteins and bacterial nucleic acid sequences. Such patterns are also expressed in abnormal cells. In this context, it is expected that nanostructured materials in the size range of proteins, protein aggregates, and viruses with different molecular coatings can engage in a sophisticated interaction with the immune system. Nanoparticles can be recognized or passed undetected by the immune system. Once detected, they can be tolerated or induce defensive (inflammatory) or anti-inflammatory responses. This paper describes the different modes of interaction between nanoparticles, especially inorganic nanoparticles, and the immune system, especially the innate immune system. This perspective should help to propose a set of selection rules for nanosafety-by-design and medical nanoparticle design.

## 1. Introduction

The immune system of higher vertebrates encompasses a collection of different specialized cells and specialized soluble molecules distributed throughout the body, being present in all organs and tissues, circulating in blood and lymph (to reach every corner of the body in case of need), and concentrated in some lymphoid organs (lymph nodes, spleen, bone marrow, where hematopoiesis takes place in adult life). These cells have been classified into two functional branches, namely innate and the adaptive immunity, which have different roles, complementing each other very efficiently in complex organisms such as mammals (simpler organisms such as invertebrates only display a perfectly efficient innate immunity). The innate immune system’s role is to scan the body to remove apoptotic bodies, cell debris, and protein aggregates; recognize and eliminate pathogens or abnormal cells; and keep commensals outside tissues. Additionally, it promotes the repair of damaged tissue and is involved in the control of embryogenesis and delivery. We can say that the innate immune system is the actual immune system, active throughout evolution with conserved and very efficient defensive mechanisms. The other system, adaptive immunity, developed much later as a complement of innate immunity, providing slower but more specific protective responses, good for long-living and mobile organisms that do not stably reside in the same environment [1]. The adaptive immune responses are tools for the innate immune system with subordinated or programmed functions—tools because they develop without making any decision [2]. It is the innate immune system that detects, categorizes, and triggers the immune response and, in the case of additional needs, calls for adaptive immunity to come in when the innate activation has reached a certain threshold level indicative of excessive danger and the need for more specific defensive tools.

These complex defensive actions that body tissues perform in response to harmful stimuli, such as pathogens or damaged cells, are described as inflammation. Inflammation requires an excess biological workout and therefore it is closely related to metabolism. Immunometabolism has become increasingly popular since the publication of Mathis and Shoelson’s perspective in 2011 [3]. This is crucial in the context of interactions with nanoparticles (NPs), since they have been observed to have the capacity to increase or decrease reactive oxygen species (ROS), which directly correlates with the onset or remission of inflammation [4]. ROS are free radical molecules resulting from natural metabolism, which, when excessive and unregulated, may contribute to cell damage and aggravate human pathologies such as cancer [5], neurodegeneration, and stroke, among others [6].

Following the great oxidation event some 2.3 billion years ago, oxidation has been the leading force of metabolism. A delicate equilibrium between heat generation (enthalpy) and biological organization (entropy) was established, which allowed natural systems to decrease their free energy in a particular controlled fashion [7]. Deregulation of a living system, for instance, in the case of a disease, increases enthalpy generation at the expense of entropy. The system over-burns, which in biological terms is described as inflammation (literally *setting in flames*). Inflammation is correlated with a particular metabolic pathway, anaerobic glycolysis, providing higher energy power output, in which cells defend themselves from aggression. Furthermore, aerobic glycolysis, with a broken Krebs cycle, provides important metabolic intermediates and ROS [8]. Inflammation provokes the unbalance between endogenous production of free radicals and antioxidant defenses, resulting in oxidative stress [9]. While this metabolic defense mechanism is an ability of all eukaryotic cells, it is reasonable to imagine that, through evolution, some cells adapted the unbalanced energy equation to becoming professional defensive cells forming a whole discontinued system distributed across the body and responsible for the maintenance, defense, and repair of our biological tissues. In normal conditions, these cells have a patrolling role based on scanning and surveying tissues to eliminate senescent or damaged cells and become aggressive when encountering some possible dangers, capable of initiating, developing, and controlling inflammation.

The innate cell response is different, depending on the type of stimulus or combination of stimuli, the stimulus intensity (quantitative and temporal), the location of the innate cells (the tissue and its specialization), and the microenvironmental conditions. All these cues trigger a defined activation profile in innate cells, which is different based on the combination of microenvironmental conditions that have triggered it. Engineered NPs may share several characteristics of microbial agents, such as size and ordered molecular surface patterns, presenting “eat-me” or “eat-me-not” surface signals that favor or prevent macrophages from engulfing them. Thus, they are expected to develop complex and intense interactions with immunity. Bachmann et al. [10] showed that the immune system readily recognized antigen repetitive organization on the surface of viral particles.

In contrast, poor antigen organization does not induce an immune response. The same holds for complement (in particular C1q, an ancient version of immunoglobulins) [11], which recognizes ordered antigenic structures as those present on microorganisms but do not react to disordered patterns as those present in mammalian cell surfaces. The same has been observed with NP coatings [12]. These interactions mainly concern innate immunity, as responsible for detecting and categorizing foreign matter inside the body.

In order to navigate the described interactions between NPs and the immune system, it is recommendable to remember the different type of immune cells and their different functions (Figure 1). A major role in the innate immune system is played by macrophages, which in mammals develop some specialized functions depending on the tissue where they reside and are named accordingly: Langerhans cells in the skin, Kupffer cells in the liver, osteoclasts in the bone, microglia in the brain. Other innate immune cells are the innate lymphoid cells (ILCs), such as natural killer cells. ILCs contribute to patrolling tissues (abundant in the barrier tissues such as mucosal surfaces), identifying and killing/eliminating abnormal cells and microorganisms, and contributing to tissue development and homeostasis. Contrary to macrophages, they cannot phagocytose, but they are endowed with cytotoxic tools that literally kill the target. Similarly, mast cells are highly efficient defensive cells, abundant in all barrier tissues, endowed with an array of pre-formed proteolytic enzymes and other bioactive substances, which they release upon challenge and can detoxify snake and bee venoms, release factors that initiate/enhance a tissue-localized protective inflammatory reaction against parasites, and contribute to tissue repair and remodeling. Other important innate cells are neutrophils (short-lived, very abundant in the blood, highly phagocytic and inflammatory, strong producers of reactive oxygen species (ROS) in response to microbes), basophils (functionally similar to mast cells but residing in the blood), and eosinophils (with partially overlapping functions with mast cells and basophils, involved in response to multicellular parasites). Moreover, cells of adaptive immunity include T and B lymphocytes, which develop membrane receptors or antibodies able to recognize different pathogenic molecules/antigens specifically. In between, there are dendritic cells, which share with macrophages the capacity of taking up, processing, and presenting pathogen-derived antigens to adaptive immune cells, thereby enabling T and B cells to develop their antigen-specific membrane receptors and antibodies.

The majority of the works on NP and immune system interactions [13] have focused on circulating blood monocytes (macrophage precursors) and tissue-resident macrophages [14]. Blood monocytes come from the bone marrow, while tissue macrophages can be a mixture of self-replicating cells that have populated the tissue during embryogenesis (developed from precursors in the yolk sac or fetal liver) and cells developed from blood monocytes that enter the tissue for replenishing the resident macrophage pool [15]. Macrophages present higher phagocytic activity than monocytes and can be easily identified based on size (monocytes are smaller than macrophages) and some biochemical markers (e.g., esterases) and surface molecules (e.g., CD14, CD16, CD68, CD11b, MAC-1) that are differentially expressed between the two cell types. The functional profile of monocytes and macrophages is exceptionally plastic, as the role of these cells is that of rapidly reacting to different microenvironmental signals by adopting an appropriate activation profile, which will contribute to danger elimination and, eventually, instructing the subsequent adaptive immune responses.

## 2. The NP-Immune System Interactions

It is essential to realize that inorganic matter is commonly nanostructured and that NPs and nanostructures have naturally occurred on the planet’s surface since its origin before life emerged. This suggests that the immune system of living organisms has developed in an environment rich in such structures and substances, and therefore it should know how to deal with them. However, since the advent of nanotechnology, we are building more artificial nanostructures and artificial combinations of nanostructures and molecules, which may result in stronger interactions with the immune system, beneficial or detrimental. They depend on the nature of the employed material and its evolution in the environment before encountering the immune cell. The observed interactions between NPs and the immune system can be classified as in Figure 2. It is essential to understand that the interaction of NPs and the immune system can be multifactorial; size, shape, and surface state (including composition, structure, charge. and hydrophilicity) are primary factors, together with the presence of bioactive moieties in the sample, or the promotion of chemical reactions resulting in immune activation. These factors are closely related to the NP evolution in different media, which may result into aggregation, dissolution, or associate with by-stander (bio)molecules. All these factors should be taken into account to develop safe NPs and functional NPs.

### 2.1. When NPs Are Not Detected by the Immune System

The first scenario is when the NPs can escape from immune system detection. Indeed, the progress in the construction of synthetic nanostructures as delivery vehicles, contrast agents, or medical devices has allowed for the development of NPs able to escape the immune system and reach their target without inducing an undesirable inflammatory reaction. In order to escape from pattern recognition receptors inserted in the cell membrane of immune cells, or opsonization by complement molecules or antibodies (which mark foreign/dead cells for recycling, thereby enhancing phagocytosis), the use of polymers to camouflage the NP surface has been thoroughly developed. In pharmacology, polyethylene glycol (PEG) and polyvinylpyrrolidone (PVP) have been the most used polymers to stealthily NPs from the immune system [13,16,17], historically developed together with the liposomal formulation of antineoplastic drugs, such as in the case of Doxil^®^ and, more recently, mRNA vaccine platforms [16]. Studies showed that these polymer functionalized NPs may appear invisible (stealth) to the immune system [17,18] by mimicking non-dangerous biological structures. As a consequence, the PEG coating increased the half-life of organic and inorganic NPs in the bloodstream from minutes to hours [19,20], similar to previous observations with PEGylated proteins [21]. Similarly, oligosaccharide- and peptide-derived NP coatings seem to afford escape from the immune system and allow for longer circulation times. Nevertheless, these coatings can reduce/delay opsonization and phagocytosis, but they do not completely prevent it. Thus, for example, the development of anti-PEG antibodies has been reported in several patients, which led to faster clearance of subsequent doses of PEG-coated formulations [22,23,24]. To circumvent these problems, researchers have explored the possibility of coating NPs with natural substances that can more finely deceive the immune system, for instance, by coating NPs with albumin or serum mixtures. As far as proteins are not denatured, and the resulting object is not too big, NPs seem to pass undetected [25].

The different coatings employed to escape immune system detection are listed in Table 1.

Another way of camouflaging NPs from immune elimination is using coatings with proteins that are downregulatory immune signals. This is the case of the CD47 protein, a marker of “self” and “eat-me-not” that is expressed on all cell membranes [34]. In the work of Rodriguez et al. [32], the attachment of “self” peptides computationally designed from human CD47 protein onto polystyrene NPs achieved a delayed macrophage-mediated clearance of NPs in mice. In addition, this increased the circulation time of the NPs and enhanced the drug delivery to lung adenocarcinoma xenografts. Likewise, Hu et al. [33] coated PLGA NPs with a red blood cell (RBC)-membrane shell. These RBC-based polymeric NPs also showed a longer circulation half-life and sustained in vivo drug release compared with that achieved by using PEG-coated NPs. The coating with specific “self” molecules can also be used for the opposite reason, i.e., to induce an immune system activation by coating NPs with endogenous danger-associated molecular patterns. As an example, Aldossari et al. [35] coated AgNPs with high-density lipoprotein (HDL), which is recognized by scavenger receptors (SR-B1) expressed by macrophages. Once administered to mice, HDL-coated AgNPs provoked the recruitment of inflammatory cells, whereas SR-B1-deficient mice showed reduced cell recruitment. This strategy allowed the antimicrobial activity of AgNPs to be enhanced by targeting delivery. This indicates how important the NP coating is to escape the immune system, where a large body of knowledge has been developed to allow NPs to serve as drug delivery vehicles. 

### 2.2. When NPs Are Detected by the Immune System and Tolerated

When the immune system detects NPs, they can be tolerated or induce an activation. Being tolerated means that NPs are silently removed, without inducing inflammation, as if they were protein aggregates, apoptotic cells, or cellular debris. This can be controlled mainly by modulating NP size, surface charge, and hydrophobicity/hydrophilicity of their surface [13,16,17,32,33], where small sizes, hydrophilicity, and negative surface charges often result in tolerable NPs [36]. In general, one can say that NPs below 4–6 nm can pass undetected and undergo rapid renal clearance after i.v. administration [37]. As the NP diameter increases, NPs become the target of the different immune cells. NPs of virus-like size (a few tens to a few hundreds of nm) can be endocytosed without triggering inflammation [12]. Larger objects, of micrometric size, like bacteria, are phagocytosed, while for sizes larger than 10–20 microns, objects are encapsulated [38].

This has critical consequences for the biodistribution of NPs inside the body. NPs are transported and accumulated in different organs depending on the administration route, their physicochemical properties, and their detection by the immune system. Accordingly, after intravenous (i.v.) injection, common NPs are often filtered in the liver by hepatocytes [17,18] or eventually Kupffer cells (the liver macrophages) depending on if they are detected or pass undetected by the immune system [31]. The first studies of biodistribution of colloidal particles (a few hundred nm) were reported in the 1970s in the *Journal of the Reticuloendothelial Society* (now *Journal of Leukocyte Biology*). Singer et al. [39] and Adlersberg et al. [40], by treating mice with i.v. and i.p. colloidal Au, found that after one hour, 90% of the administered dose was accumulated in the liver and 10% in the rest of the body (mainly kidneys). Subsequent histological studies with similar colloidal gold particles i.p. administered found them in the liver and lymph nodes primarily localized inside macrophages [41]. These results were later confirmed by numerous studies of the pharmacokinetic and biodistribution of different NPs. Sadauskas et al. [42], using AuNPs of different sizes (below 40 nm), showed that Kupffer cells were central in accumulating NPs once they entered the body. Similar results were also obtained with metal oxides, quantum dots, carbon nanostructures, etc. [43]. Yokel et al. [44] administered citrate capped nanoceria (5, 15, 30, and 55 nm) at 50 and 100 mg/kg bw i.v. into Sprague-Dawley rats and measured Ce content over time (1 h, 20 h, and 30 days). Remarkably, in all these works and many others, no inflammation or systemic injury was observed, except at larger doses (>100 mg/kg bw). Accordingly, we have observed by mass spectroscopy that after i.v. administration of albumin-conjugated nanoceria (CeO_2_) at low doses (0.1 mg/kg bw), twice a week during two weeks, in control and fibrotic Wistar rats, that most of the Ce is in the liver (84% of the administered dose after one hour and 75% eight weeks after administration) [26].

This non-inflammatory capture of NPs can be exploited for harnessing these phagocytic immune cells to transport NPs towards the target area, be it a wound, an infection, or a tumor. For such delivery, circulating monocytes have been proposed as a sort of Trojan Horse or Cellular Shuttle, since they naturally migrate from the blood to the sites of damage and disease. Hence, they can be loaded with NPs to be transported through the body [45,46]. Thus, Choi et al. [47] explored the use of monocytes containing AuNPs for transport into tumor regions for subsequent photothermal therapy. This study showed the phagocytosis of AuNPs by both monocytes and macrophages and their recruitment into the tumor. Oude-Engberink et al. [48] showed the accumulation of monocytes laden with iron oxide NPs (30 nm) in the affected cerebral sites in a rat model of experimental neuroinflammation. More recently, Moore et al. [49], using a microfluidic in vitro model, showed increased activity of monocytes/macrophages to transport NP across a confluent endothelial cell layer, advancing in the design of cellular shuttles loaded with NPs. This tolerated elimination of NPs may limit the dispersibility of NPs inside the body. However, the immune system is by itself an important therapeutic target where nanocarriers can efficiently transport drugs assisting immunotherapy.

### 2.3. When NPs Are Detected by the Immune System and Not Tolerated

Many reports show that NPs may induce harmful immune responses and toxicity. NPs can induce an inflammatory immune activation because of aggregation or dissolution or because they accidentally carry immune-activating moieties (such as endotoxin, detergents, allergens, or cationic molecules). These biological effects are rather independent of the composition, size, or shape of the individual NP, described as extrinsic factors of NP toxicity [4]. Similarly, the organization of molecular epitopes in a non-conventional form (upon adhesion of the NP surface) may generate new antigens or allergens. The activation of the immune system induced by NPs can be classified as follows:

#### 2.3.1. NP-Induced Oxidative Stress

The more universal inflammatory reaction to NPs corresponds to the most non-specific and rapid defense mechanism of macrophages, the overproduction of reactive oxygen species (ROS), which results in oxidative stress, responsible of lipid oxidation and DNA damage and, eventually, structural alterations, DNA mutations, and cell death. ROS refers to biogenic free radical molecules resulting from natural metabolism characterized by being highly oxidant. These free radicals are involved in different critical physiological processes, such as gene expression, signal transduction, growth regulation, and, significantly, inflammation, where high ROS concentrations are needed to sustain the energetic demands of a proinflammatory immune response [50].

Accordingly, independently of composition, large aggregates of TiO_2_ [51], Al_2_O_3_ [52]_,_ and Fe_2_O_3_ [53] NPs showed a similar capacity to increase oxidative stress. Moreover, the corrosion process of metallic NPs itself produces a high concentration of free radicals, which may trigger an inflammatory immune response [54,55,56]. These processes are often neglected in NPs made up of bulk non-biodegradable materials. However, biodegradation of Ag, Fe_3_O_4_, and CdSe/ZnS NPs due to enzymatic or hydrolytic activities in lysosomes [57,58] have been described. Even the physiological disintegration of AuNPs through oxidative etching by cysteine and chlorine has been described [59,60,61]. Similarly, carbon nanotubes (CNTs) have been observed to dissolve in vivo through enzymatic catalysis [62]. Subsequently, an increased number of reports has established relationships between observed inflammatory effects after NP exposure and NP disintegration [63,64,65,66]. Related to that, it is worth mentioning the works of Burello et al. [67] and Zhang et al. [55]. They developed theoretical and experimental models to predict the oxidative stress potential of oxide NPs by looking at their bandgap energy and their ability to perturb the intracellular redox state. Note that NP dissolution may become a source of toxic cations. For instance, in the early 2000s, the studies of Derfus et al. [68] and Kirchner et al. [69] showed that the released Cd ions were responsible for the intracellular oxidation and toxicity caused by CdSe NPs. Similar effects were found later when comparing the toxicity of Ni NP and ions as a function of time [70].

#### 2.3.2. When Phagocytosis Is Not Sufficient

When the immune system detects a foreign object, phagocytosis is the first mechanism for elimination that comes into play. However, when the object is too big (usually larger than 10–20 μm), rather than engulfing it, the immune cells start spreading on it to form a layer of cells that secludes the object from the rest of the tissue and initiate a chemical defense against the material that, if not non-biodegradable, is permanently kept secluded into a fibrous capsule or granuloma.

Historically, chronic inflammation has been observed in the case of penetration of non-biodegradable (persistent) large size (micrometric) particles in the lungs, as the well-known cases of particle-induced granulomatosis such as silicosis and asbestosis [71]. This is because when a phagocytic cell fails to digest these particles, phagolysosomal rupture, the release of lysosomal enzymes and particles, and subsequent activation of the inflammasome and other cytoplasmic sensing mechanisms may happen, thereby triggering inflammation. This may lead, as the material persists, to chronic inflammation, permanent oxidative stress, tissue damage, and alterations that favor tumorigenesis in the long term. Brandwood et al. [72] found that murine macrophages phagocytosed inert carbon fiber-reinforced carbon particles up to 20 microns in diameter, but larger particles were not engulfed and became surrounded by aggregations of macrophages. This reaction may have pathological aspects; fibromas and granulomas are non-functional neo-tissues, similar to scars, that may hamper the organ functions and be active, i.e., growing, for a long time. In some instances, the reaction can be overtly pathological, as in the case of long fibrous materials. Accordingly, when Poland et al. [73] instilled high doses of multiwalled carbon nanotubes between the membranes lining the lungs and abdominal organs in mice, they found that long straight nanotubes caused inflammation and lesions in membrane cells similar to those leading to cancer, just like asbestos fibers [74]. Similarly, Ag nanospheres did not elicit any immune response or toxicity, while Ag nanowires can elicit a high inflammatory response, directly correlated to nanowire length, in murine macrophages [75]. The same effects were observed by Ji et al. [76] in THP-1 cells when comparing nanoceria nanorods and high aspect ratio nanoceria nanowires at high doses and aggregation states. This suggests that the needle-like shape of NPs is prone to provoke inflammation. It has been observed that macrophages engulfing needle-shaped crystals and fibers end up getting pierced by the needle-like structures and, consequently, start inflammation [77]. Parental NPs usually are never grown to these sizes, but uncontrolled aggregation can transform objects of tens of nm to tens of μm.

#### 2.3.3. NPs, Intendedly or Accidentally, Can Display Antigens, Allergens, or Toxins

The unintended or accidental absorption of biomolecules onto the NP surface may be a cause of concern. NPs may associate with specific bio-molecules, toxic by-standers, or pollutants, and present them to the immune system in an ordered pattern, thereby mimicking microorganisms and triggering the innate immune reaction of the host.

It is important to note that NPs have a strong tendency to adsorb many different molecules (hetero-aggregation) at their surface due to their high surface energy. Consequently, they are usually surrounded by a molecular coating, either provided intentionally (NP functionalization) and/or spontaneously by molecules present in the environment, forming the NP biocorona. These coatings also take part in the NP morphology and functions. The consequences of this are diverse; NPs can be good molecular aggregators and substrates for molecules to be presented to the immune system.

Among essential immunoactive biomolecules present in the environment, bacterial endotoxin is one of the most common and abundant. Endotoxins (also known as lipopolysaccharides (LPSs)) are large molecules present in the outer membrane of Gram-negative bacteria, able to elicit strong innate/inflammatory immune responses. Endotoxin is a ubiquitous environmental contaminant and can be present in all chemicals and glassware used in laboratories, even after sterilization (depyrogenation is needed to get rid of it). The presence of endotoxin, if not recognized, can be responsible for many of the in vitro and in vivo effects attributed to NPs [78]. Our study [79] showed that the endotoxin present on AuNPs turned those NPs from inactive to highly inflammatory and able to induce secretion of IL-1β in human primary monocytes. This could be an underlying factor in inflammatory responses and toxic effects associated with other metallic NPs and carbon nanomaterials [80,81]. Hence, special attention is needed to avoid endotoxin contamination when preparing NPs, which includes working in endotoxin-free conditions and glassware depyrogenation [82].

In other cases, the toxic ingredient may come from the formulation or derived chemicals employed during NPs preparation. If the synthesis process does not involve proper purification steps, the use of such NP samples may entail deleterious responses due to excess surfactants or unreacted precursors. This is the case of PEI molecules, a common NP stabilizer to enhance NP endocytosis, but with safety concerns due to the attachment to the negatively charged cell membranes that modify permeability and compromise viability [83]. Indeed, it is well-established that positively charged macromolecules can cause higher toxicity and immune activation than their neutral or negatively charged counterparts, as in the case of monolayer-coated silicon nanoparticles [84]. Similar is the case of amphipathic molecules such as cetyltrimethyl ammonium bromide (CTAB), employed in preparing Au nanorods [85], which act as a detergent to lyse cell membranes. Another example is in the work of Dowding et al. [86]. These authors prepared different nanoceria NPs using the identical precursor (cerium nitrate hexahydrate) through a similar wet chemical process but using other bases: NH_4_OH, which yields negatively charged nanoceria, or hexamethylenetetramine (HMTA), which yields positively charged nanoceria at neutral pH. Results showed that HMTA-nanoceria NPs were readily taken into endothelial cells and reduced cell viability at a 10-fold lower concentration than the other NPs, which showed no toxicity.

Another type of immunoactive molecule that NPs can adsorb is allergens. This is unlikely to happen in the case of NPs since the concentration of allergens in the environment is very low, and the NP surface would be passivated before encountering them. However, it must be taken into account. Note that it has been reported that car combustion emission microparticles, when coated by pollen grains, enhance allergenic responses [87]. Radauer et al. [88] observed the formation of a stable allergen coating around NPs when exposed to different types of allergens (Der p1 and Bet v1), enhancing allergic responses against them. A recent review about the potential of NPs to trigger allergies via adsorption of allergens can be found in reference [89]. Here, it is essential to remark that allergy, understood as an anomalous immune response towards substances that are generally tolerated, has never been observed for engineered nanomaterials per se.

Regarding immune effects induced by biomolecules adsorbed on the NP surface, another possible source of inflammation comes from the potential modification of the structure of proteins upon adsorption at the NP surface [90]. Lynch et al. [91] pointed out how partial protein misfolding at the NP surface may result in the exposure of protein fractions usually buried in the core of the native structure. These cryptic epitopes may be recognized by immune cells and trigger inappropriate defensive reactions. Accordingly, in the work of Falagan [92], such modifications of the adsorbed proteins structure have been indicated as responsible for the long-term toxicity observed after a single low-dose exposure of AuNPs.

#### 2.3.4. NPs Presenting Vaccine Antigens and Working as Vaccine Adjuvants

Regarding the intentional use of NPs for vaccination, conjugation of antigens to NPs can help both attain a more efficient presentation of poorly immunogenic soluble antigens and provide an adjuvant effect targeted to innate immunity (the NP as a foreign agent) [12]. An interesting example is the development of AuNP-based virus-like particles (VLPs), where the NP replaces the virus core, which scaffolds the proteic capsid structure [93]. Typically, capsid proteins need the highly negatively charged dense core of DNA/RNA to self-assemble properly. This core can be replaced by dense and highly negatively charged AuNPs. Nikura et al. [93] demonstrated that the size and shape of AuNP-VLPs allowed for shaping of the in vitro and in vivo immune response in terms of the production of antibodies against West Nile virus. This implies that by modulating the NP size and shape, and consequently the arrangement of viral proteins on the NP surface, it could be possible to obtain highly effective and efficient vaccines. NPs can also be employed as vaccine adjuvants by exploiting their capacity to target and modulate the activity of innate immune cells. For a long time, vaccines were prepared by precipitation of antigens within some matrix, initially bread crumbs (in the XIX century), and currently alum powder, where the antigens are absorbed, forming aggregates that vary in size from 1 to 20 μm, acting as an antigen depot [94]. In this way, slow release of antigens is achieved, prolonging antigen presence, improving its processing and presentation. Other NP aggregates have been used as adjuvants. Skarastina et al. [95] used silica NPs (10–20 nm) as adjuvants for the hepatitis B vaccine in a mouse model. The monodisperse silica NPs formed heterogeneous aggregates larger than 1 μm after formulation, resulting in the same IgG2a/IgG1 ratios as in the case of immunization with alum as an adjuvant. Other nanostructure used as vaccine adjuvants are nano-sized emulsions (sometimes called lipidic NPs). This is the case of the oil-in-water MF59 emulsion, which is used as an adjuvant, mainly for influenza vaccines (Flaud^®^, Novartis) and has been licensed in more than 20 countries. The MF59 adjuvant allows for significant cross-reactivity against viral strains and reduces antigen concentration to 50–200-fold lower doses [96].

The induction of inflammatory responses with non-pathogenic triggers has also been proposed as a preventive approach against exposure to unknown pathogens. Behind this concept, preventive activation of the innate immunity, there is the capacity of innate immune cells to develop a different response to a challenge as a consequence of previous contact with a different threat, a capacity known as innate memory [97,98]. All innate immune cells are able to develop a long lasting memory, despite their short lifespan in circulation. The reason lies mainly on the fact that the precursors in the bone marrow can be primed by a trigger and generate circulating immune cells with a different capacity to react against threats, as currently shown in monocytes/macrophages. Thus, after having previously experienced an inflammatory activation, the innate immune system becomes more efficient in preventing the rooting of newly incoming pathogens. For instance, this has been observed in the case of the administration of bacille calmette-guerin (BCG), the vaccine for tuberculosis, which increases resistance to other diseases [99]. The generation of innate memory thus represents an alternative, or better a complement, to the highly specific adaptive memory induced by vaccines. The strategy of innate memory induction leads to outcomes (enhanced protection) that have advantages (wider range of protection) and disadvantages (more unpredictable and less controllable side effects). Despite the controversy regarding safety, NPs can be used as adjuvants for the non-specific amplification of immune responses, and, even more, they can be excellent tools for generating or modulating innate memory [100]. In this regard, administration of AuNPs alone was observed to have little/no impact on the subsequent capacity of human monocytes to mount an innate/inflammatory response to a microbial challenge (LPS) [101]. However, the co-administration of AuNPs, or Fe_3_O_4_NPs, with memory-inducing microbial agents (e.g., LPS, BCG, muramyl dipeptide (MDP), Helicobacter pylori) led to a modulation of the innate memory response induced by the microbial agents depending on the priming stimulus and the NP type, shape, and size [102]. The implication is that vaccination with antigens and NPs could bring about a protective specific immunity based on adaptive immune memory and a non-specific innate memory induced by the antigen-NP combination.

The proinflammatory activation effects are listed in Table 2. In all these aspects, the uncontrolled proinflammatory activation of the immune system is, in principle, a common source of NP toxicity. In contrast, controlled activation can be employed for vaccination and other modes of defense against pathogens.

### 2.4. When NPs Act as Enzymes and in This Way Can Modulate Immune Reactions

Rare-earth oxide NPs have been found to be biocompatible antioxidants able to buffer excess ROS in physiological conditions, showing powerful anti-inflammatory effects. Mineral antioxidants may offer superior activity to currently available substances due to their enhanced bioavailability and stability, longer tissue residence time, and resistance to biological degradation [106]. These features can be exploited in many diseases based on excessive immune/inflammatory activation, such as autoimmune diseases, chronic inflammation, organ rejection, asthma and other allergic diseases, neurodegenerative pathologies (Alzheimer’s disease, Parkinson’s disease), and aging [106]. They have been described as engineered inorganic materials with enzyme-like activities, especially cerium oxide NPs, nanoceria [106,107]. Nanoceria has been reported to display superoxide dismutase (SOD)-like activity (conversion of superoxide anion into hydrogen peroxide and finally oxygen) [108], catalase-like activity (conversion of hydrogen peroxide into oxygen and water) [109,110], peroxidase-like activity (conversion of hydrogen peroxide into hydroxyl radicals) [111], as well as NO scavenging ability [103]. Consequently, nanoceria has been shown to safely down-regulate oxidative stress by scavenging the excess of ROS in diseases such as retinal degeneration [112,113], neurological disorders (including Alzheimer’s disease, Parkinson’s disease, and ALS) [114,115,116], ischemia [117], cardiopathies [118], diabetes [119], gastrointestinal inflammation [120], liver diseases [26,27,28], and cancer [121,122], as well as in regenerative medicine [123] and tissue engineering [124], with better performance than other antioxidant substances in both efficacy and efficiency. Interestingly, nanoceria become active at high ROS concentrations. Otherwise, at homeostatic ROS levels, the NPs become inactive. This is because several free radicals have to be simultaneously absorbed onto the NP surface in order to be recombined into non-radical adducts, a condition that only happens for high ROS concentrations. In other words, the ROS scavenging capacities of nanoceria are ROS concentration dependent. With time, these NPs dissolve into innocuous ions, which are excreted via the urinary route [19]. The solid NPs have been observed to be excreted through the hepatobiliary route [26,125].

This aspect is significantly different from the previous ones, where activation of the immune system results in inflammatory responses. In this case, the enzyme-like catalytic activity of rare earth NPs results in anti-inflammatory activity. The different observed responses are listed in Table 3.

## 3. NP Evolution and Transformations in the Exposure Media

The interaction between NPs and the immune system strongly depends on the conditions in which such interaction occurs (route of exposure, co-exposure with other agents) and on the characteristics of both NPs and the host immune system. Here we have presented the variety of immune responses to NPs and how these responses can help us design immuno-active and immune-benign NPs, which could either avoid immune recognition and activation in order to persist in the body long enough for completing their theranostic tasks or directly interact with immune cells for triggering inflammatory or anti-inflammatory responses as desired for therapeutic purposes. Indeed, the scientific community is still struggling with the apparent contradiction of similar materials being toxic and non-toxic (even beneficial) at the same time. This paradox can be attributed to undescribed effects of NP modifications during their dispersion in the working media, such as aggregation and corrosion. The main modifications NPs may suffer during their dispersion in different media are shown in Figure 3. Basically, NPs can be coated with molecules (e.g., hydrophilic polymers) to both pass undetected by the immune system and avoid aggregation (1). They can also be coated with soluble antigenic molecules to induce a response against them (2). In the opposite direction, when dispersed in physiological media, NPs can aggregate (3) and adsorb other molecules present in the medium (e.g., protein corona) (4) or both (5). In addition, depending on the core composition, NPs can be used as ROS scavengers, thereby down-regulating inflammatory responses (6), or they can dissolve and act as an ion reservoir that may increase the level of oxidative stress and generate an inflammatory response (7).

NPs have different ways to minimize their high surface energy, basically aggregation and corrosion. These are common phenomena in nature, widely studied by geochemistry, where a NP is an intermediate state between a micrometric particle and the dissolved ions. Thus, NPs may aggregate or associate with coating molecules in different media. They may also disintegrate through corrosion (defined as the chemical degradation of a solid material) and dissolution.

Aggregation deserves particular attention. NPs are colloidally stable by repulsive electrostatic or steric forces or a combination of both. Aggregability depends on intrinsic NP parameters such as morphology, surface coating, and charge, and extrinsic parameters such as electrolyte concentration, pH, presence of organic matter, etc. Aggregation is common in physiological environments where NPs aggregate to submicrometric or micrometric sizes when not properly stabilized. To avoid it, one has to provide repulsive forces to the NP surface, either by electrostatic (high surface charge) or steric (entropic) means, usually provided by large soluble molecules associated with the NP surface; otherwise, they will aggregate and their unique physicochemical properties that arise at the nanoscale (quantum confinement, superparamagnetism, extreme catalytic activity, etc.) progressively lost. Note that aggregation entails modifications in terms of specific surface area, concentration, mobility, and dosing. Protein adsorption, the formation of a protein corona, is a particular aggregation phenomenon between NPs and proteins present in the dispersion media. It is a dynamic process in which, initially, proteins adsorb and desorb at the surface, followed by a set of re-organizational arrangements, which make this absorption more stable and finally irreversible [126,127]. This depends on NP size, surface state, type of protein, and protein-NP incubation media and conditions, where sophisticated functional patterns can be obtained [128]. The most straightforward strategy to cope with this issue is to passivate the NP surface in a controlled manner, e.g., by albuminization, PEGylation, or addition of PVP. These strategies usually decrease aggregation, even in high salt media, and the adsorption of microenvironmental biomolecules on the NP surface.

In addition to aggregation, chemical transformations, corrosion, and dissolution can also be a cause of immune activation via the alteration for the cellular redoxome, and the delivery of toxic cations. In this regard, the nanochemist or nanoengineer needs to control the redox potential (and the oxidative/reductive environment) where the NP will be stored, employed, and disposed of. In this regard, using NPs at their higher valence state is recommendable when possible [14] (passivating the surface with a continuous layer of oxide is sometimes an alternative). The chemical transformation and dissolution of NPs, which can cause immune activation or toxicity, is fundamental to determine NP fate and reduce its presence and persistence in the environment.

## 4. Concluding Remarks

After considering the different interactions NPs may have with the immune system, one can draw indications on how NPs have to be designed to control these interactions and consequent responses. Indeed, while many NP functions can be attributed to their core structure, the surface coating defines much of their bioactivity. By controlling the nano-bio interface, NPs can be designed to be safe and innocuous or active and have therapeutic benefits. From the NP point of view, and for the nanochemist and the nanoengineer, the NP immunological properties can be summarized as depicted in Figure 4. The composition of the NP core determines its chemical potential and catalytic activity, while the surface coating largely determines its bioactivity. NPs can aggregate, either with other NPs or with macromolecules present in the physiological media (e.g., biocorona), or they can dissolve, being redox-active and acting as an ion reservoir, consequently increasing the levels of oxidative stress.

In order to address the NP immune interactions, one has first to deal with the instability of the NP surface, and it should be passivated before introduction into biological systems. Otherwise, they may spontaneously aggregate, resulting in objects of increased size and decreased dose. Polymeric coatings have traditionally been the most commonly employed materials for such purposes. However, this surface engineering is sometimes costly and involves multi-step synthesis approaches, sometimes in the organic phase. One simple and effective solution could be to promote NP solubility in physiological media by pre-albuminization during the preparation process [118,119,128], a similar approach employed by Abraxane^®^, one of the first approved nanomedicines [30]. In addition to providing colloidal stability and avoiding opsonization, NP coatings can be designed to directly interact with the immune system, such as CD47 [32] for avoiding complement activation, LPS for inducing innate/inflammatory activation [79], or viral proteins for vaccination [95]. Finally, NPs that belong to the family of natural antioxidants, such as nanoceria that catalytically scavenge free radicals (ROS in the context of inflammation), provide powerful immunomodulatory effects.

Thus, by mainly playing with surface characteristics, it is possible to adjust the NP physicochemical characteristics (aggregation, surface display of given biomolecules, chemical stability) and consequently their modes of interaction with the immune system.

## Figures and Tables

**Figure 1 nanomaterials-11-02991-f001:**
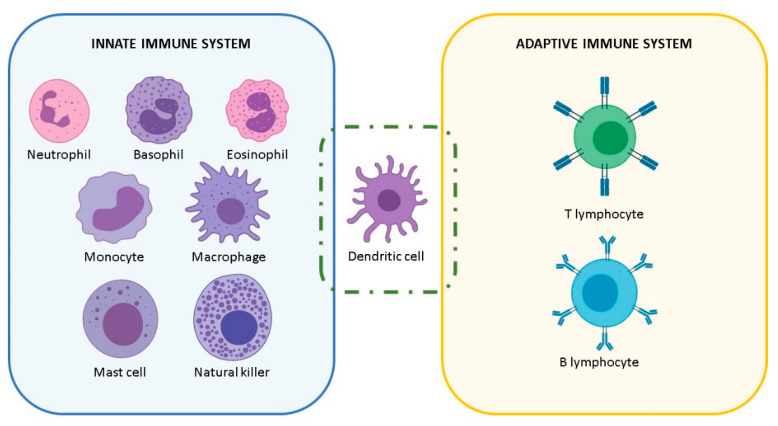
Representative classification of the most common mammalian immune cells.

**Figure 2 nanomaterials-11-02991-f002:**
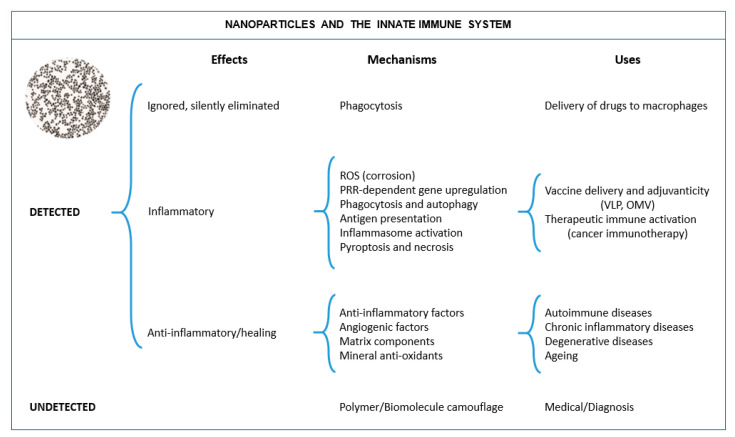
NP-immune system interactions. NPs can be undetected or detected by cells of the immune system, depending on different parameters such as their size, surface charge, and hydrophobicity/hydrophilicity of the surface coating. If detected, NPs can be either tolerated (either ignored or eliminated in a silent fashion, i.e., without inducing an inflammatory reaction) or generate an inflammatory response allowing for resolution of inflammation and tissue regeneration or an anti-inflammatory. With a proper NP design, these responses can be harnessed for developing different immunomodulating activities for medical exploitation (e.g., self-adjuvanted vaccines based on virus-like particles (VLPs), or outer membrane vesicles (OMVs), etc.).

**Figure 3 nanomaterials-11-02991-f003:**
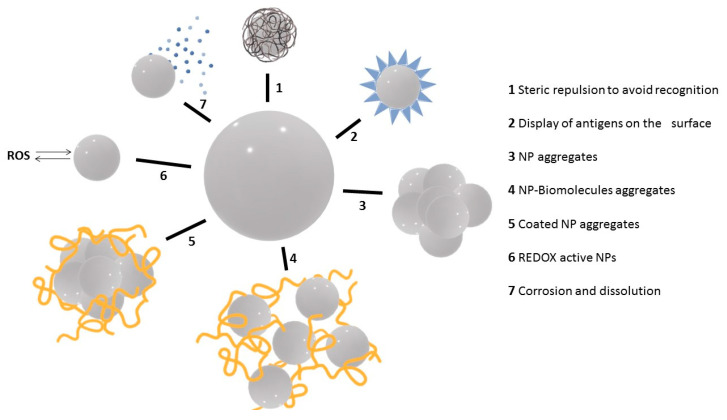
Intended or spontaneous NP modifications and their impact on immune responses.

**Figure 4 nanomaterials-11-02991-f004:**
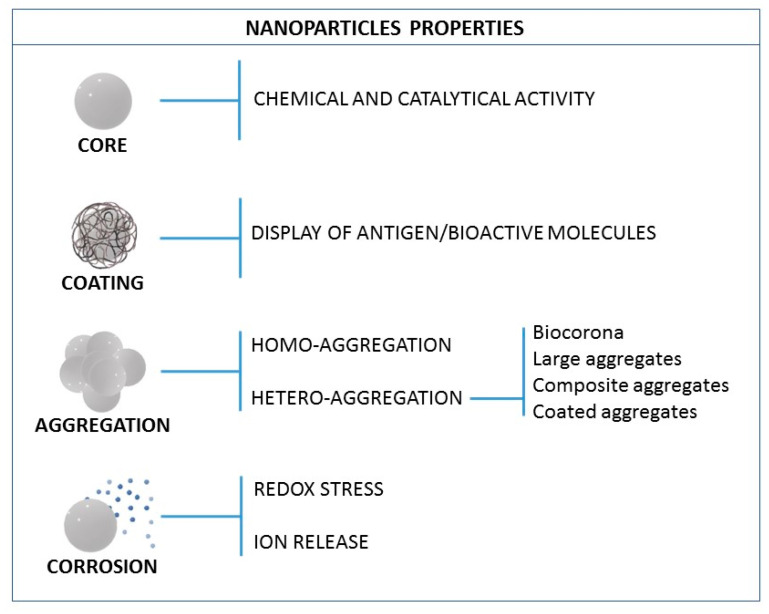
The NP properties that can impact the immune system.

**Table 1 nanomaterials-11-02991-t001:** Coatings employed to avoid immune detection.

Surface Coating	Nanoparticle Core	References
Albumin (rat, mouse, or human)	CeO_2_ NPs as an anti-inflammatory mineral substance	[26,27,28,29]
	Abraxane as an albumin-based nanoparticle for chemotherapeutic delivery	[30]
Polyethylene glycol (PEG)	Au NPs for tumor targeting	[19]
	SiO_2_ NPs for evasion of phagocytic clearance	[20]
	Polystyrene NPs for evasion of phagocytic clearance	[31]
Retinol	Polyethylenimine (PEI) NPs for drug delivery	[25]
CD47	Polystyrene NPs for evasion from phagocytic clearance	[31,32]
Erythrocyte membrane fragments	Poly(lactic-co-glycolic acid) (PLGA) NPs for drug delivery	[33]

**Table 2 nanomaterials-11-02991-t002:** Inflammatory activation induced by NPs.

NPs That Cause Inflammation		
Category	Surfactant	References
Inflammation induced by by-standers	Cetyltrimethyl ammonium bromide (CTAB)	[85]
	Hexamethylenetetramine (HMT)	[103]
Inflammation induced by pollutants	Bacterial endotoxin (LPS)	[78,79]
	Allergens	[88,89]
**Category**	**Mechanism**	**References**
Inflammation induced by the core	Non biocompatible size/shape	[72,73,74,75,76]
	Excess of aggregation/agglomeration	[51,52,53,96,102,104]
	Chemical transformations and corrosion	[54,55,56,57,58,59,60,61,62,63,64,65,66,67,68,69,70,102]
**Category**	**Surfactant**	**References**
Inflammation induced by the coating (bioactive molecules, VLPs…)	Virus like particles (VLP)	[93]
	Antigen/ordered peptides/proteins coatings	[91,92,100,105]
	Cationic polymers	[83]

**Table 3 nanomaterials-11-02991-t003:** Immune responses to NP exposure.

Category	Nanoparticle Core	Surface Coating	References
NPs that pass unnoticed	Au NPs	Polyethylene glycol (PEG)	[19]
	SiO_2_ NPs	Polyethylene glycol (PEG)	[20]
	Polyethylenimine (PEI) NPs	Retinol	[25]
	Polystyrene NPs	CD47 or PEG	[31,32]
		Bovine serum albumin (BSA)	[46]
	Polymeric NPs	Erythrocyte membrane fragments	[33]
	Abraxane	Human serum albumin (HSA)	[30]
NPs that are tolerated	Au NPs	Sodium citrate	[42,92]
		Disordered peptidic coatings	[47]
	CeO_2_ NPs	Rat serum albumin (RSA)	[44]
	Polystyrene NPs	Poly-L-lysine	[46]
	Fe_3_O_4_ NPs	Dextrane	[48]
	SiO_2_ NPs	3-Aminopropyltriethoxysilane (APTES)	[49]
Immunoactive NPs with inflammatory activity	Au NPs	Peptides/proteins	[55,58,93,105]
		Bacterial endotoxin (LPS)	[78]
		Cetyltrimethyl ammonium bromide (CTAB)	[85]
		Allergens	[88]
		Poly(acrylic acid) (PAA)	[92]
		Polyethylene glycol (PEG)	[92]
	Ag NPs	High-density lipoprotein (HDL)	[35]
		Sodium citrate	[58,65,74]
	Alumina NPs	Fetal bovine serum (FBS)	[52]
	CeO_2_ NPs	Hexamethylenetetramine (HMT)	[86]
		Polyethylenimine (PEI)-polyethylene glycol (PEG)	[122]
	Gadolinium endohedral metallofullerenols	Polyhydroxy	[80]
	Silica NPs	Hepatitis B virus core protein	[104]
Immunoactive NPs with anti- inflammatory activity	CeO_2_ NPs	Murine serum albumin	[26,27,28,29]
		Polyethylene glycol (PEG)	[117]
		Gelatin	[124]

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
