# Peer review of "The Interactions between Nanoparticles and the Innate Immune System from a Nanotechnologist Perspective"

_nanomaterials, 2021, doi:10.3390/nano11112991_

Round 1
Reviewer 1 Report
This review article outlines various interaction modes on how the innate immune system recognize nanoparticles and discusses their applications further. In general, the topic is of course interesting and important considering their potential in nanomedicine design. However, there has published a number of review articles summarising the interactions between nanoparticles and the innate immune system. Compared to others, this review at the current form fails to provide new and precise understandings to the community. Thus, I recommend the publication of this manuscript after major revisions, my detailed comments are as follows.
1- In the introduction section, nearly four pages of context systematically summarized the backgrounds of the immune system, but there are only two references being cited, is this reasonable? I suggest authors have a careful check of related references being mentioned but not properly cited. The same issue occurs in the discussion and future directions sections. For example, “using NPs at their higher valence state is recommendable but not always possible. Passivating the surface with a continuous layer of oxide is sometimes an alternative. Besides, the presence of crystal impurities may favor NPs dissolution, as in the case of TiO2” Do these recommendations do not have relevant literature support?
2- The layout of these sections and the way to organize them need to be properly revised, and some descriptions are unclear to the readers. For example, in the next section “When NPs are not detected by the immune system” the author only mentioned “This can be controlled mainly by modulating their size, surface charge, and hydrophobicity/hydrophilicity conferred by their surface coating.” It would be good to show more evidence to support this conclusion. “Different studies showed that these polymer functionalized NPs may appear invisible to the immune system.” The reason for invisible is missing. “When NPs are detected by the immune system and tolerated section” author mentioned refer 37 provide more mechanistic insights with a lack of further discussion. “When NPs are detected by the immune system and not tolerated section” the author only simply mentioned “Aggregability depends on intrinsic NP parameters” but without any detailed explanation. In addition, some explanation seems not consistent with the previous conclusion. For instance, the authors mentioned natural substances coating is useful to prevent NPs from immune system detection, but subsequent examples explained not only the charge but also the small size resulted in non-detection. Moreover, I cannot catch up on some cases connected in a paragraph, which are not relevant to each other. For instance, the author mentioned pollutants during NPs preparation may cause immune response but then listed NP preparation under different bases and made a conclusion that positively charged nanoceria showed no toxicity.
3- Although the authors summarized various types of interaction modes, the underlying mechanism of this phenomenon is not properly discussed. For example, the authors mentioned the phenomenon of NPs detected by the immune system and even gave a definition of tolerated and not tolerated. However, the reason behind this phenomenon should be addressed, e.g. under which situation and why the immune system is tolerated.
Author Response
Dear Editor, we are thankful for the reviewer comments and how they will contribute to improve our work. Below, a detailed description of the corrections and explanations for the reviewers. The written english has been also reviewed.
REVIEWER 1
(x) I would not like to sign my review report
( ) I would like to sign my review report
English language and style
( ) Extensive editing of English language and style required
( ) Moderate English changes required
(x) English language and style are fine/minor spell check required
( ) I don't feel qualified to judge about the English language and style
Comments and Suggestions for Authors
This review article outlines various interaction modes on how the innate immune system recognize nanoparticles and discusses their applications further. In general, the topic is of course interesting and important considering their potential in nanomedicine design. However, there has published a number of review articles summarizing the interactions between nanoparticles and the innate immune system. Compared to others, this review at the current form fails to provide new and precise understandings to the community. Thus, I recommend the publication of this manuscript after major revisions, my detailed comments are as follows.
We would like to highlight that in contrast to other existing reviews, this focus on the NP features that trigger the different immune responses, in terms that can be designed and modified by nanoengineers developing nanomaterials. This “from the NP point of view” should bring closer the main selection rules to develop safe NPs to non-biologists, specially nanochemists. Despite that, we acknowledge that there are other great reviews on the subject and we have included them in the manuscript.
1- In the introduction section, nearly four pages of context systematically summarized the backgrounds of the immune system, but there are only two references being cited, is this reasonable?
However we would like also to note that the introduction is a brief description of what we consider is the minimal description of the immune system for nanotechnologists, full of basic concepts available in academic text books, which not need to be referenced. Note that the paper already contains about 130 refs, including new references to support the description of the immune system in the introduction (2 to 9).
I suggest authors have a careful check of related references being mentioned but not properly cited. The same issue occurs in the discussion and future directions sections. For example, “using NPs at their higher valence state is recommendable but not always possible. Passivating the surface with a continuous layer of oxide is sometimes an alternative. Besides, the presence of crystal impurities may favor NPs dissolution, as in the case of TiO2” Do these recommendations do not have relevant literature support?
We have referenced were these point is addressed. Indeed we would say that are well-known things in solid-state material science, and likely NANOMATERIALS readers, where oxidized materials are more stable than non-oxidized and where crystal impurities weaken the crystal structure.
2- The layout of these sections and the way to organize them need to be properly revised, and some descriptions are unclear to the readers. For example, in the next section “When NPs are not detected by the immune system” the author only mentioned “This can be controlled mainly by modulating their size, surface charge, and hydrophobicity/hydrophilicity conferred by their surface coating.” It would be good to show more evidence to support this conclusion. “Different studies showed that these polymer functionalized NPs may appear invisible to the immune system.” The reason for invisible is missing.
We regret that this point was not clear in the text, where we have made small changes to direct the reader to the numerous references given on the paper about this aspect. Coatings mimicking non-danger biological structures as in the case of functionalization with serum albumin, Polysaccharides, PEG or PVP... as described in references 10 to 12, and 20 to 28.
“When NPs are detected by the immune system and tolerated section” author mentioned refer 37 provide more mechanistic insights with a lack of further discussion. “
We have deswcribed the contribution of ref.37 (now 44).
More recently, Moore et al.[44], using a microfluidic in vitro model, showed an increased activity of monocytes/macrophages to transport NP across a confluent endothelial cell layer when they were exposed to a physiological flow compared to static conditions, advancing in the design of cellular shuttles loaded with NPs for therapeutic intervention.
When NPs are detected by the immune system and not tolerated section” the author only simply mentioned “Aggregability depends on intrinsic NP parameters” but without any detailed explanation.
This has been explained followingly:
Aggregability depends on intrinsic NP parameters such as morphology, surface coating, and charge, and extrinsic parameters such as electrolite concentration, pH, presence of organic matter, etc. NPs often tend to reduce their high surface energy by aggregation with other NPs or any encountered matter. To avoid it, one has to provide repulsive forces to NP surfaces, either by electrostatic means (high surface charge) or steric (entropic) means, normally provided by large soluble molecules associated to the NP surface, otherwise they will aggregate andtheir unique physicochemical properties that arise at the nanoscale (quantum confinement, superparamagnetism, extreme catalytic activity, etc.) will be progressively lost.
In addition, some explanation seems not consistent with the previous conclusion. For instance, the authors mentioned natural substances coating is useful to prevent NPs from immune system detection, but subsequent examples explained not only the charge but also the small size resulted in non-detection.
We regret we did not made our selves clear, detection by the immune system can be multifactorial where size and surface state are important factors. This has been included in the text, in a new paragraph following also reviewer 2 comments (vide infra).
Moreover, I cannot catch up on some cases connected in a paragraph, which are not relevant to each other. For instance, the author mentioned pollutants during NPs preparation may cause immune response … but then listed NP preparation under different bases and made a conclusion that positively charged nanoceria showed no toxicity.
Clearly, the origin of the immune response can be related as much as the presence of contaminants as the surface charge, specially if it remains positive after dispersion in physiological media.
We have clearly separate the two points, in the first case it is an unintended ingredient of the NP formulation while in the second case it is an active ingredient of the formulation. In the second case it should not be called pollutant, what probably confounded the reader… we have fixed and clarified this point.
Besides, in this section nanoceria with positive charges DO SHOW toxicity.
3- Although the authors summarized various types of interaction modes, the underlying mechanism of this phenomenon is not properly discussed. For example, the authors mentioned the phenomenon of NPs detected by the immune system and even gave a definition of tolerated and not tolerated. However, the reason behind this phenomenon should be addressed, e.g. under which situation and why the immune system is tolerated.
This has been described in the precedent responses, where mimicking non-danger signals makes tolerable NPs.
Reviewer 2 Report
In my opinion this paper is very interesting as it nicely describes different aspects of NP-immune system interactions. However, certain aspects should be more stressed - such as the role of material, size and shape of NPs. The NPs differ in type of material, type of conjugation/coating, the manner of coating (covalent/non-covalent) and this determines their immune properties. This should be well explained at the beginning instead of the end of mnuscript (Figure 4).
There are also minor corrections:
Line 87 - Other innate immune cells are the Innate Lymphoid Cells, cells of lymphoid 86 origin but with functions similar to tissue macrophages (the best known are Natural Killer 87 -NK- cells). – this statement is unclear – macrophages without phagocyting proterties (as wrtitten in next line).
Line 199 – I disagree with statement that small NPs, below 10 nm can pass undetected by the immune system. Small size unables to directly cross the lipid bilayer of cells and cause toxicity, relase of cellular content and induction of inflammation.
Line 594 – In my opinion, Figre 4 should be placed at the begining of the manuscript. The type, shape, size of NPs decides about the imunological activities. Coating can be attached covalently or non-covalently, which strongly impacts durability of the immune stimulation.
Author Response
REVIEWER2
English language and style
( ) Extensive editing of English language and style required
(x) Moderate English changes required
( ) English language and style are fine/minor spell check required
( ) I don't feel qualified to judge about the English language and style
In my opinion, this paper is very interesting as it nicely describes different aspects of NP-immune system interactions. However, certain aspects should be more stressed - such as the role of material, size and shape of NPs. The NPs differ in type of material, type of conjugation/coating, the manner of coating (covalent/non-covalent) and this determines their immune properties. This should be well explained at the beginning instead of the end of mnuscript (Figure 4).
There are also minor corrections:
We agree that it would help the reader to include the information on the NP evolution and the different NP derivates presentation to the immune system at the beginning of the NP-Immune system interactions. However, we prefer not to move backwards figure 4 since it also helps the reader, once the different observations described, to extract selection rules for the design of immunosafe NPs.
The added paragraph reads as follow:
We regret we did not made our selves clear, detection by the immune system can be multifactorial where size and surface state are important factors. This has been included in the text, in a new paragraph following also reviewer 2 comments (vide infra).
The interaction of NPs and the immune system is multifactorial were size and surface state (which would include composition, structure, charge and hydrophilicity) are main factors together with the presence of bioactive moieties or the promotion of chemical reactions which results in immune activation. These factors are closely related to the NP evolution in different media. When NPs are removed from their synthesis bath and dispersed in their working media they may aggregate, dissolve, associate with biomolecules present in the media. All this factors should be taken into account to develop safe NPs or to design them to selectively interact with the immune system, as in the case of antigen presentation.
Line 87 - Other innate immune cells are the Innate Lymphoid Cells, cells of lymphoid 86 origin but with functions similar to tissue macrophages (the best known are Natural Killer 87 -NK- cells). – this statement is unclear – macrophages without phagocyting proterties (as wrtitten in next line).
This has been corrected.
Line 199 – I disagree with statement that small NPs, below 10 nm can pass undetected by the immune system. Small size enables to directly cross the lipid bilayer of cells and cause toxicity, relase of cellular content and induction of inflammation
This has been removed.
Line 594 – In my opinion, Figre 4 should be placed at the begining of the manuscript. The type, shape, size of NPs decides about the imunological activities.
We thank the comment to help the reader describing the NP different parameters and transformations which may result in immune activation. We however still thing that moving figure 4 backwards would not help while adding a new paragraph (vide supra) on this subject make the paper more readable.
Reviewer 3 Report
Dear Authors,
This is a timely piece of literature. However the manuscript could be better presented.
- For each section please summarise with table indicating studies and outcomes.
- NPs need more work in terms of differences between lipid, polymer, biomimetic, microvesicle NPs etc and there uses in innate and adaptive immunity, inflammation etc. Use tables and diagrams
Author Response
REVIEWER3
( ) Extensive editing of English language and style required
( ) Moderate English changes required
(x) English language and style are fine/minor spell check required
( ) I don't feel qualified to judge about the English language and style
Dear Authors,
This is a timely piece of literature. However the manuscript could be better presented.
- For each section please summarise with table indicating studies and outcomes.
We thank that for the table is already figure 2. However, following the reviewer suggestion we have summarized conclusions at the end of each section:
This indicates how important is the coating of the NP to escape the immune system, where a large body of knowledge has been developed to allow NPs to serve as drug delivery vehicles.
More recently, Moore et al.[44], using a microfluidic in vitro model, showed increased activity of monocytes/macrophages to transport NP across a confluent endothelial cell layer when they were exposed to a physiological flow compared to static conditions, advancing in the design of cellular shuttles loaded with NPs. This tolerated elimination of NPs may limit the dispersibility of NPs inside the body, however, the immune system is an important therapeutic target where nanocarriers can efficiently transport drugs assisting and enabling immunotherapy.
In all these aspects, the uncontrolled pro-inflammatory activation of the immune system is in principle a common source of NP toxicity why under control, it can be employed for vaccination and other modes of defense against pathogens.
This aspect is significantly different from the previous ones where activation of the immune system results in pro-inflammatory responses. In this case, the enzyme-like catalytic activity of rare earth NPs results in anti-inflammatory activity.
2. NPs need more work in terms of differences between lipid, polymer, biomimetic, microvesicle NPs etc and there uses in innate and adaptive immunity, inflammation etc. Use tables and diagrams
Reviewer is right. Despite that many of the aspects applies to either organic or inorganic NPs, the paper is focused on inorganic NPs. This has been specified in the abstract and the intro.
Round 2
Reviewer 1 Report
I thank the authors to provide a response to my comments, but unfortunately, I was disappointed to see the authors were reluctant to reshape this manuscript into a better quality through major revisions to meet the high standard of this journal. ALL of my comments were dealt with not much care. Especially for the response to my 1st comment regarding the long but known knowledge, "full of basic concepts available in academic text book, which not need to be referenced." I have to say if such a large part of this manuscript comes from a textbook, this manuscript does not deserve to be published in a journal again. A review paper summarises the current status of research, not to convey textbook knowledge, which of course could be organized and summarised, but in brief! Following that response, the subjective presumptions from the authors also made this manuscript being not very scientific.
Author Response
Following reviewer suggestions, the text has been revised thoroughly. Concepts have been organized and presented in a more order form, references removed and new more appropriate added, the context better described and interlinked with the subject of the paper, the descriptions have been improved, the text severely reorganized, superfluous sentences, references and paragraphs removed, the structure of the paper improved and the descriptions will categorized and explained in their section (before some concepts were scattered across de whole text redundantly, what leads to confusion).
This is strongly related to the format of the paper, the contents have been kept. This are important concepts important for many nanotechnologist with rather a chemistry or physics background than life sciences. We have been working on the subjects for decades in International consortium and Diana Boraschi is a well known senior expert on innate immunity.
A version with highlighted changes is uploaded in this section for your perusal.

Reviewer 2 Report
No further comments
Author Response
thanks, following other reviewers comments we have reorganized the paper and included tables,
Reviewer 3 Report
Dear Author,
Thank you for making modifications/clarifications to the article. However, as this is a review article i do believe readers will benefit if there were tables related to studies and outcomes
Author Response
Following reviewer suggestion, tables listing the different categories of coating, inflammatory response and immune response described in the scientific literature have been included.